# Hippocampal neuronal representations in continual learning

## Abstract

The hippocampus has long been associated with spatial memory and goal-directed spatial navigation. However, the region's independent role in continual learning of navigational strategies has seldom been investigated. Here we analyse population-level activity of hippocampal CA1 neurons in the context of continual learning of two different spatial navigation strategies. Demixed Principal Component Analysis (dPCA) is applied on neuronal recordings from 612 hippocampal CA1 neurons of rodents learning to perform allocentric and egocentric spatial tasks. The components uncovered using dPCA from the firing activity reveal that hippocampal neurons encode relevant task variables such decisions, navigational strategies and reward location. We compare this hippocampal features with standard reinforcement learning algorithms, highlighting similarities and differences. Finally, we demonstrate that a standard deep reinforcement learning model achieves similar average performance when compared to animal learning, but fails to mimic animals during task switching. Overall, our results gives insights into how the hippocampus solves reinforced spatial continual learning, and puts forward a framework to explicitly compare biological and machine learning during spatial continual learning.

## 1 Introduction

The hippocampus has been long known to play a crucial role in spatial navigation (O'keefe & Nadel (1978); Tolman (1948)). However, there is still an ongoing debate in the field on how exactly the hippocampus is involved in the encoding and learning of multiple tasks, such as allocentric and egocentric spatial tasks (Ekstrom et al. (2014); Feigenbaum & Rolls (1991); Fidalgo & Martin (2016)). In particular, it is not clear how the hippocampal neural population may encode such tasks and how these representations relate to the ability of animals for continual spatial learning.

In reinforcement learning (RL), despite the growing interest in the topic, how to solve the continual learning problem is still unresolved (Parisi et al. (2019)). Previous theoretical work has explored how animals may rely on the hippocampus to learn to explore a given environment based by combining RL algorithms and spatial coding (Foster et al. (2000)). More recently, there has been growing interest in understanding how to solve continual learning in deep neural networks (Zenke et al. (2017); Kirkpatrick et al. (2017)). But how both biological and artificial neural networks encode and solve spatial tasks in a continual reinforcement learning setting is not known.

Here, we first focus on analysing neural data from hippocampus CA1 population-level activity to extract meaningful latent features. We find representations that are directly related to how the neural population encodes not only multiple tasks, but also reward signals, such as reward prediction errors. Next, we contrast these experimental observations with standard reinforcement learning algorithms, namely temporal difference learning and Q-learning theory. Finally, we show that deep Q networks can achieve an average performance similar to animals, but fails to achieve the same relearning speed.

## 2 Results

Our results are based on a dataset of neural activity from a total of 612 hippocampal CA1 neurons, recorded during a reward-based T-maze traversal task (Ciocchi et al. (2015)). For the behavioural

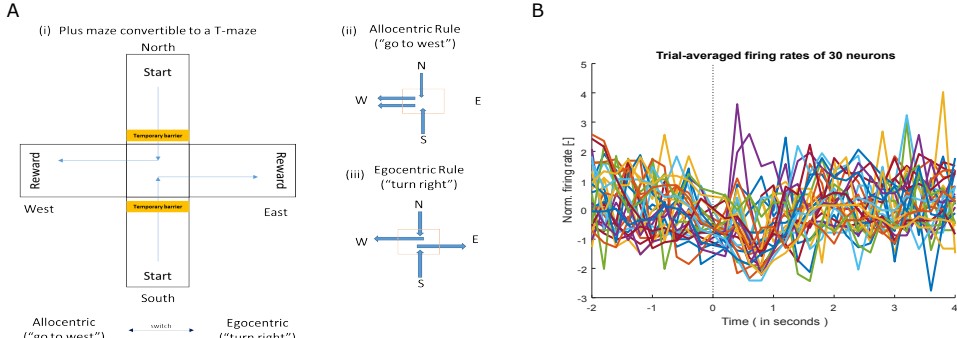

Figure 1: Experimental setup and neuronal data. (**A**) Experimental task setup. (i) The experimental results are based on a plus maze which can be converted to a T-maze by temporarily blocking the opposite start arm (yellow box) after pseudo-random selection of a *start* location during an allocentric or egocentric spatial task. There are two possible start points (North and South) and two reward arms (West and East). During a training session the rules (or tasks) are switched multiple times. (ii) Representation of the allocentric task in which the animal has to navigate the west arm where a reward is located. (iii) Representation of the egocentric task in which the animal has to turn right to get a reward irrespective of start point. Figure adapted from Ciocchi et al. (2015). (**B**) Example of trial-averaged z-scored firing rates of 30 randomly selected CA1 neurons out of 612 around the reward point (vertical dashed line).

experiment five food-deprived rats were trained to get sugar pellet rewards located at the extremities of rewarded arms in a T-maze (Fig. 1Ai). Rats were trained to learn two behavioral rules in order to be rewarded. The first is based on allocentric spatial references (Fig. 1Aii) and the second on egocentric spatial references (Fig. 1Aiii). We analyse a total of 6 seconds of neuronal activity per trial across several days (around 500 trials in total). For illustration, trial-averaged and normalized firing activity of 30 randomly selected neurons from the recorded CA1 ensemble is presented in Fig. 1B. From the raw firing rates is it virtually impossible to uncover meaningful trends around the reward delivery zone (dotted vertical line). As a result, we decided to focus on population-level analysis of the neuronal activity.

Recently a task-specific linear dimensionality reduction method, Demixed Principal Component Analysis (dPCA), has been introduced. This algorithm has previously been tested on population-level data from the prefrontal and orbitofrontal cortices by Kobak et al. (Kobak et al. (2016)), but not on data from hippocampal spatial tasks. Next, we describe our results obtained using dPCA and how these may be related to reinforcement learning algorithms.

## 2.1 BEHAVIOUR-SPECIFIC COMPONENTS IN CA1 NEURONAL POPULATIONS

We use demixed PCA to extract meaningful features from the data. Like PCA, dPCA (Kobak et al. (2016)) allows us to extract a lower dimensional representation of the data, but in contrast to PCA in dPCA this representation is constrained such that the components extracted encode behaviourally relevant features (e.g. spatial strategy or correct vs incorrect decisions; see Supplementary Material for more details on dPCA). The dPCA components extracted from the hippocampal CA1 data explained above are given in Figure 2.

Although individual neurons do not exhibit a clear tuning for decision (i.e. correct or incorrect trials) or strategy (i.e. allocentric or egocentric tasks; cf. Fig. 1B), at the population level we observe a clear separation of the neural representations for decisions and strategies (Fig. 2B). Interestingly, we observe a clear separation in population activity patterns for correct and incorrect decisions in the leading demixed decision component (Fig. 2B, first row, component 3). It is interesting to note that the separation between correct and incorrect decisions increases gradually over time. This increased separation starts before the reward delivery and increases substantially afterwards, being at its maximum where we believe to be the moment during which animals consume the actual reward (i.e. about a second of crossing the reward sensor represented by a vertical dashed line). This

reward-representation appears to be related but also different from standard reward prediction errors predicted by RL algorithms (see more below) and classical observations in dopaminergic neurons (Schultz et al. (1997); Wirth et al. (2009)).

In terms of multi-task representation, we find a clean separation at the population-level between allocentric and egocentric tasks (Fig. 2B, second row, component 4, suggesting that the hippocampus is directly involved in the learning and representation of these two tasks.

Temporal components capture the most variance in the data and their population activities reveal reward-specific or reward-approach-specific responses (Fig. 2B, third row). We interpret the first temporal component (1) as being analogous to an eligibility trace, the second temporal component (2) as being encoding reward location and the third temporal component (5) as exhibiting traces of reward prediction error signals as seen in classical conditioning experiments (Schultz et al. (1997); Fig. 2B, third row). Note that the first decision component (3) may be also interpreted as some form of (weak) reward prediction error as discussed above.

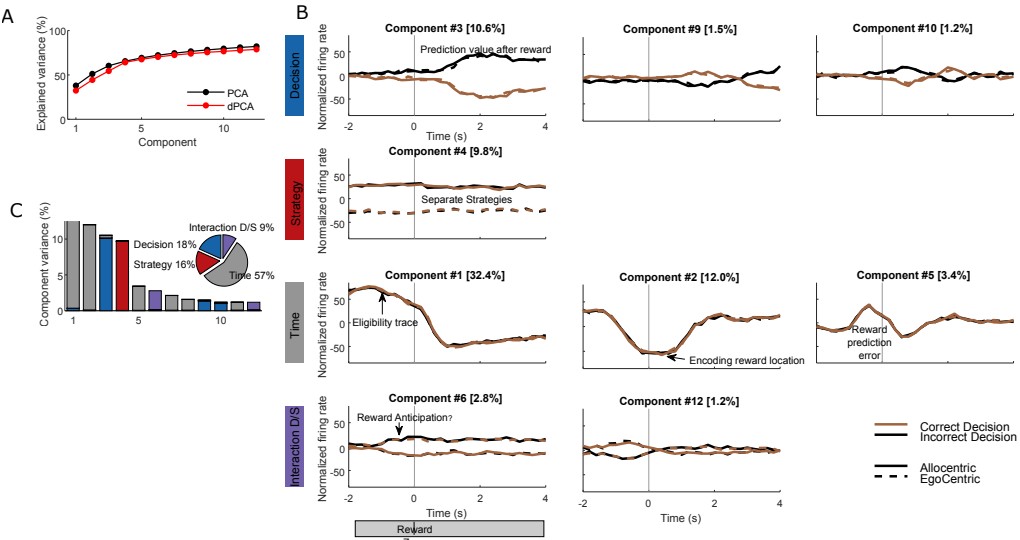

Figure 2: Demixed principal component analysis (dPCA) reveals hidden features from CA1 hippocampal recordings. (**A**) Cumulative variance explained by dPCA and PCA components. (**B**) Demixed components for each experimental variable: decision (incorrect in black and correct in brown), strategy (allocentric in solid line and egocentric in dashed line), time and mixed decision/strategy. Vertical dashed line represents the crossing of the reward sensor. We expect that the actual reward to be consumed ∼1 afterwards. (**C**) The variance explained by each demixed principal component and total variance per experimental condition (pie chart). Figure follows same structure as in the original dPCA (Kobak et al. (2016)).

These population features we observe should become better defined over learning. That is consistent with what we find when comparing earlier trials with later trials (Fig. 3). For example, the separation between the two tasks (egocentric vs allocentric) increases in later trials. A similar trend is observed between correct and incorrect trials. Finally, the eligibility trace-like component also becomes more pronounced in later trials.

## 2.2 COMPARISON BETWEEN STANDARD RL ALGORITHMS AND HIPPOCAMPAL REPRESENTATIONS

Here we contrast the features observed in the hippocampus with ideas commonly found in RL theories.

We first use a simple temporal difference (TD) learning algorithm with eligibility traces in a 1D environment (see SM for more details). Presence of eligibility traces postulated by TD models is similar to the ones we found in our experimental data (Fig. 4Ai,Bi). Regarding evidence for TD

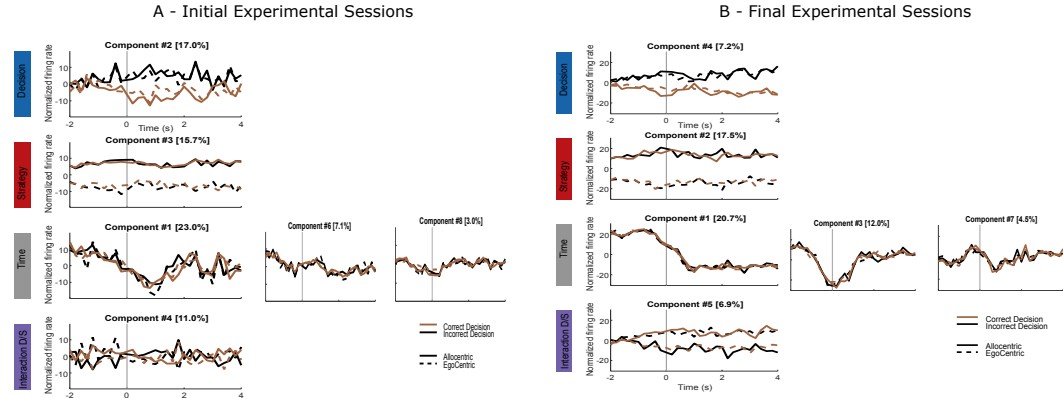

Figure 3: Demixed PCA for earlier and later training sessions. The first decision, strategy and interaction components along with the first three temporal components are plotted from dPCA on neuronal firing rates from the initial (**A**) and final (**B**) experimental sessions. The components from later days are cleaner and highlight the influence of learning the tasks over time. Vertical dashed line represents the crossing of the reward sensor.

errors, our results only loosely support standard TD errors, in that the 3rd component distinguishes between correct and incorrect trials (Fig. 4Aii,Bii). However, from Fig. 3 we can see that this component is initially stronger (i.e. in earlier trials) and becomes weaker with learning around the expected reward location (i.e. 1s after crossing the sensor), consistent with changes in TD errors over learning.

Next we used $epsilon$-greedy Q-learning in a grid-world setting to model the experimental task (see SM for details) we observe a separation between the two tasks in the value function (i.e. allocentric and egocentric; Fig. 4C,D), which appears to be similar to the one observed experimentally. However, as expected, this is not sufficient for the model to learn the two tasks in a continual learning setting as we highlight next.

Indeed, our Q-learning model fails to learn the navigational strategies consecutively (Fig. 5B) in contrast to animals that can sustain a performance above chance level (Fig. 5A). We show the performance of animal SC03 to demonstrate that animals can learn both allocentric and egocentric strategies as is noticeable from the mean performance for both tasks being above chance level. The model, on the other hand, is unable to switch over and recall policies learned for the allocentric task at trials 212-216. This is because, as expected, the policy information pertaining to the tasks is overridden as tasks or starting locations change over. This clearly highlights the continual forgetting in standard RL systems. However, it should be noted that some forgetting can be observed in animals as well (Fig. 5A).

The incapability of standard Tabular Q-learning algorithm to reach a performance similar to animals lead us to develop a deep Q-Network that may be able to better solve the two tasks.

## 2.3 COMPARING DEEP Q-NETWORKS AND ANIMAL PERFORMANCE

We implement a deep Q network (DQN) with task-specific information provided to the neural network as input. Such task-specific information to the DQN allows the network to learn the different rules of reaching reward locations and this method draws inspiration from the works of Anonymous et al. and Serra et al. (Authors; Serrà et al. (2018)). In addition, there is evidence that specific neuromodulators (Acetylcholine) may provide such task-specific information. Nevertheless, unlike earlier works our DQN does not require a separate memory layer to store the task-specific information corresponding to the different tasks. Salient task-specific information for inclusion in the blocked cells of the input maze matrix are chosen with the help of a trial and error mechanism for training the network with different positive and negative values of information. We used the first 250 trials from our animal data (Fig. 5) to train and test the network ability to learn and remember the different tasks. We used Adam optimizer to train the network.

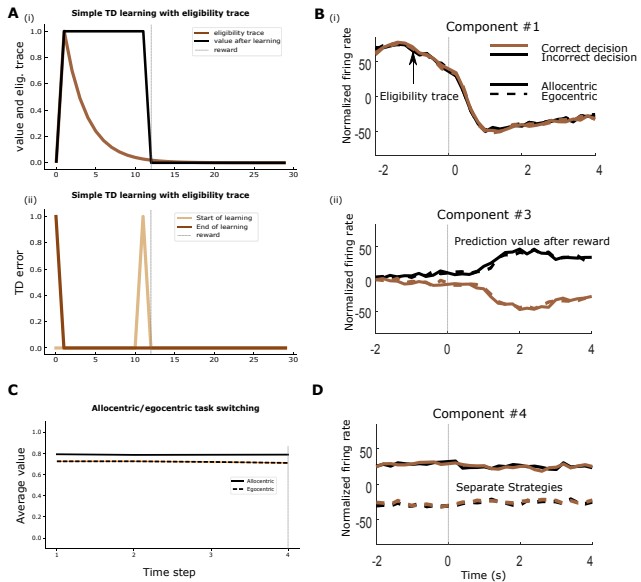

Figure 4: Comparing RL simulation results with low-dimensional dPCA components. (Ai) Value and eligibility trace from a TD($\lambda$) model simulated for a 1D environment. (Bi) Firing activity pattern in dPCA component 1 is akin to the simulated eligibility trace pattern. (Aii) TD errors before (light brown) and after learning (dark brown) from a TD($\lambda$) model simulation. The model uses a look-ahead strategy for error estimation. (Bii) Weak reward prediction signals around the reward delivery point. Error prediction signals signalling trial outcomes after reward point. (C) Separation of strategies based on Q-value estimates. (D) Distinct firing activity in dPCA component 4 pertaining to allocentric and egocentric tasks.

The DQN consists of two hidden layers of 100 neuron units each. The input layer is of the size of the input maze, while the output layer contains four neurons with the Q-values for the four different actions that an agent can take to change its state in the maze. Additionally, we found PRelu to be the most suitable activation function to solve this reinforced spatial navigation task involving allocentric and egocentric navigational strategies.

Interestingly, we observe that this network with task-specific memory achieves a performance for both allocentric and egocentric tasks comparable animals (animal SC03 is given as a typical example (Fig 5A). In particular, the DQN learns the egocentric task more efficiently than the allocentric task as demonstrated by the higher mean performance (dashed blue) line in Fig. 6. Even though the network struggles with the allocentric, it shows a mean performance above the chance level. It should also be noted that the DQN performs better in the egocentric task, while the animal finds it slightly more difficult to learn the egocentric spatial navigation task rules. In other words, animal SC03 performs marginally better in the allocentric tasks than in the egocentric tasks (Fig. 5A).

We compared our results from this DQN with task-specific information with a DQN without task-specific information and this network performed significantly better than the other one (results not shown), whose performance accuracies for both allocentric and egocentric tasks were below the chance level.

Mean performance across all Q-learning models including deep Q-networks are compared in the following Fig. 7A against the mean performance of animal SC03 over learning the allocentric and egocentric tasks for 250 trials. To highlight the ability of the different systems for continual learning we plotted the re-learning performance of the Q-learning models and the animals after the networks/models and the animal has already been exposed to the two different task types (Fig. 7(B,C)).

A DQN that learns both tasks consecutively as the animal does in the behavioural experiment achieves a mean performance comparable to the animal (green and orange bars in Fig. 7A). A DQN learning only learning the egocentric task but with a reversal of reward locations (see SM Fig. 8A(i,iv)) performs better (peach bar) than the animal. On the other hand, a DQN learning only

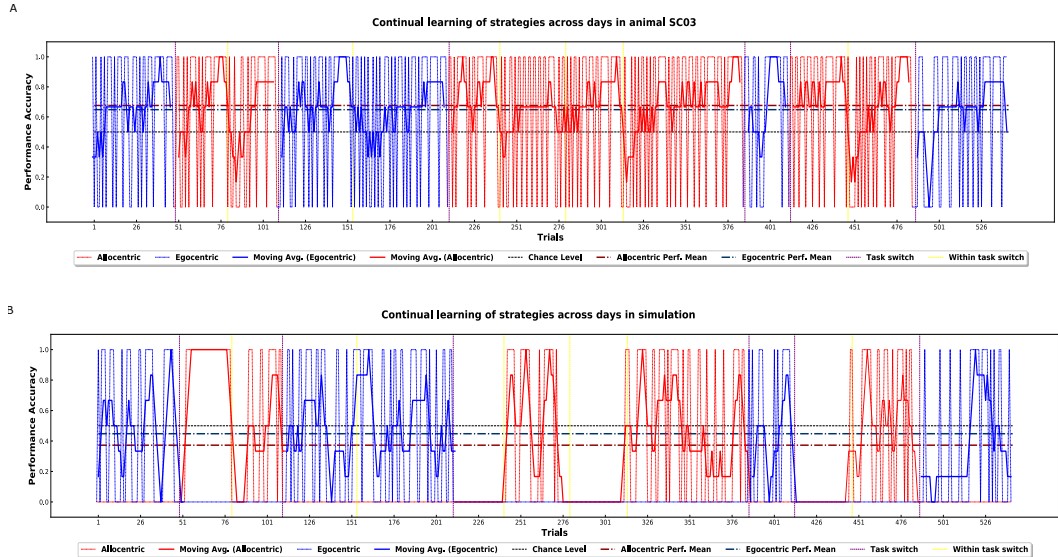

Figure 5: Behavioural performance in animals and Q-learning model during task switching. (A) Performance accuracy for animal SC03. (B) Performance accuracy for simulated model. A performance of 1 represents correct and 0 incorrect for both egocentric (blue) and allocentric (red). Solid darker lines represent moving averages. Abrupt changes in both model and animal occur within a given strategy because of changes in the start location (e.g. around the 80th trial). Color-coded vertical dashed lines indicate moments of task switches. Dashed yellow lines denote a change of reward locations while following a particular strategy. Dashed purple lines denote the change from allocentric to egocentric task or vice versa.

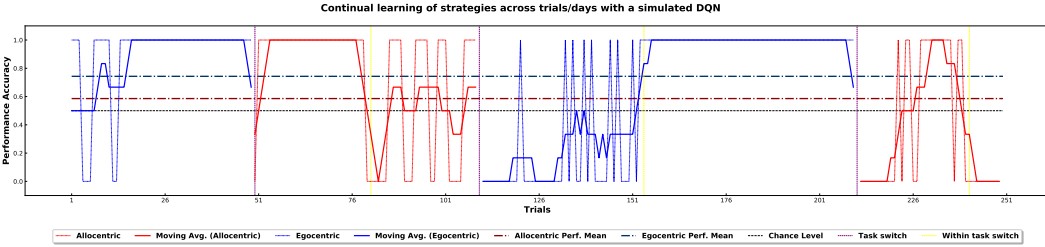

Figure 6: Task switching simulation using a deep Q network. Solid darker lines represent moving averages. Task-specific information is added in the input maze to facilitate differentiation of the allocentric and egocentric rules by the network. Mean performance in both tasks is above the chance level. The neural network achieves improved continual learning of both tasks.

the allocentric task with reward location reversals fails to acquire a performance close to both the DQN-Ego and the animal (peru colored bar). Tabular Q-learning model shows the worst mean performance amongst all the models as well against the animal's mean performance. This is expected because a standard Tabular Q-learning model is inefficient in incorporating and sustaining complete information corresponding to the different strategies.

Additionally, we speculate that the DQN-Ego acquires a better performance than the DQN-Allo since the DQN-Ego encounters far less reward location change overs (dashed 'Strategy Switch' lines) when compared to the DQN-Allo model. Providing task-specific information in the DQN also shows a clear improvement in continual learning of tasks. We postulate that the DQNs somehow learn to utilize this extra information (memory weights for tasks), perhaps internally segregating the activity patterns (even weights) corresponding to the task types. In contrast, a Tabular Q-learning model without any extra information about the tasks struggles to reach comparable performance as the animal does. This also leads to the hypothesis that the hippocampus CA1 might also be using

similar mechanism to encode the rules involved with the allocentric and egocentric navigational strategies and consequently, remember the reward locations as the task change over trials.

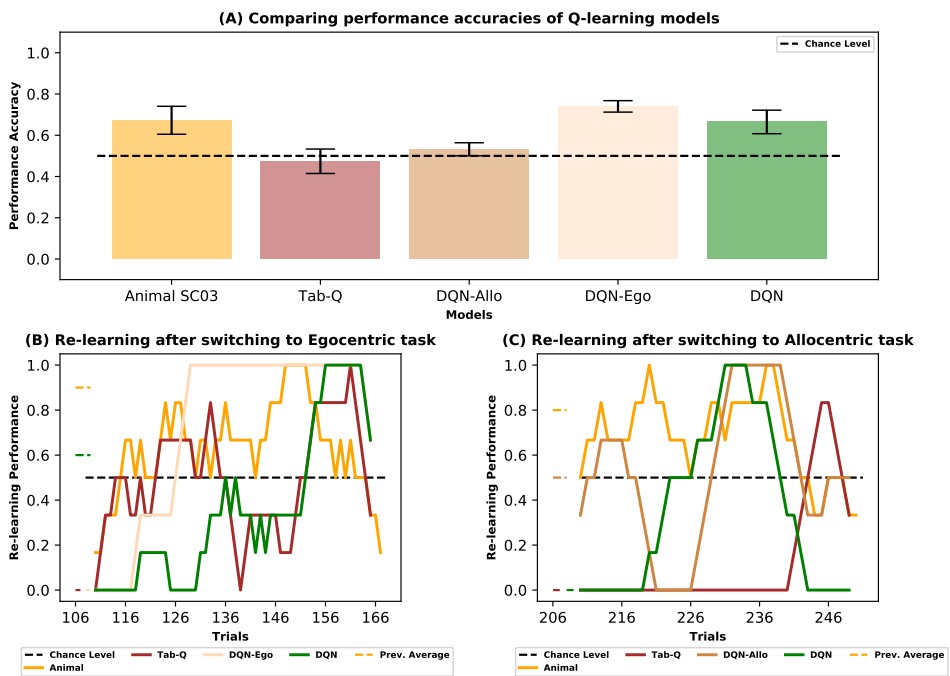

Figure 7: Comparing mean performance of all Q-learning model simulations with the animal's mean performance. Re-learning performance on allocentric and egocentric tasks are also plotted from the Q-learning models and the animal. Color-coded dashed lines at trials 106-109 and 206-209 display average performance of the networks and animal in the last few trials of the previous strategy it was learning. (A) Even the animal is unable to achieve perfect learning of both tasks (orange bar). A DQN learning both tasks achieves mean performance almost at par with the animal. A DQN learning only the egocentric task shows better mean performance opposite to the animal SC03's actual behaviour. Mean performance of a DQN model learning only the allocentric task is superior to a Tabular Q-learning model approximating optimal policies for both task types. Error bars represent the standard error of the mean for each model. (B) Re-learning performance of each Q-learning model and the animal at the point of reintroduction of a previously learned egocentric task. The re-learning speeds are color-coded similar to the bar plots. The animal has the best re-learning speed (yellow), while the DQN takes a while to catch up (green). (C) Re-learning performance of each Q-learning model and the animal at the point of reintroduction of a previously learned allocentric task. The animal has an above chance level re-learning performance, but the DQN shows a slow re-learning speed. Even the animal starts from a better prior re-learning performance value, while the DQN does not show accurate prediction immediately after the switch.

Performance at re-learning the tasks in each Q-learning model and the animal is plotted against trials after a task type is switched to another in Fig. 7(B,C). The animal starts with a higher re-learning performance value than any of the models at both allocentric and egocentric task switch points. It also achieves rapidly relearns the switched-to task and this is observed from the orange line values above chance in Fig. 7B and Fig. 7C. The DQN learning both tasks together shows a slow re-learning speed and fails to immediately re-learn the switched-to task. However, this network on average attains a performance close to the animal as can be noticed from comparing the green and orange bars in Fig. 7A for the switch to the egocentric task (brown line in Fig. 7B) than when the task is changed to allocentric rule (brown line in Fig. 7B). Networks learning only egocentric (DQN-Ego in Fig. 7B) and allocentric (DQN-Allo in Fig. 7C) have better re-learning speeds than the DQN learning both together. However, the re-learning performance for both DQN-Allo and DQN-Ego is inconsistent across trials.

Overall, standard deep RL models can achieve a performance comparable to animals, but fail to continually learn. Although animals relearn faster, they are still not perfect relearners.

## 3 CONCLUSION AND DISCUSSION

Overall, our results using dPCA show that the hippocampus encodes task variables such as decisions, rewards and strategies, which suggests that the hippocampus plays a key role in the ability for animals to perform continual spatial learning. We also demonstrate that some of the components we find are consistent with theoretical models in reinforcement learning. Our results suggest the existence of a non-standard form of TD learning implemented or used by the hippocampus. This raises exciting possibilities for future research. One avenue is to develop a model that gradually learns to separate the representations of the two tasks inspired by recent developments in continual learning where different groups of neurons may specialize in different tasks explaining the features and the ability for continual learning that we observe in experimental neuroscience (Zenke et al. (2017); Kirkpatrick et al. (2017)). Our work also puts forward a framework that can be used to better compare how both machines and animals learn to solve continual learning problems.

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

## A SUPPLEMENTARY MATERIAL

## METHODS OVERVIEW

### A.1 DEMIXED PRINCIPAL COMPONENT ANALYSIS

Demixed Principal Component Analysis (dPCA) (Kobak et al. (2016)) not only reduces the dimensionality of the data without losing too much information, but also facilitates visualization of neural

responses tuning to experimental parameters like decisions, rewards and navigation rules. Another advantage of demixed PCA over standard Principal Component Analysis (PCA) methods is that it does not enforce orthogonality of the components and hence, enables successful demixing of population-level activity. We applied the recent version of the demixed PCA algorithm Kobak et al. (2016) on trial-averaged neuronal firing rates from the hippocampus CA1 of all five rats across all experimental days. The trial-averaged data was organised into a matrix with dimensions Neurons x Decision x Strategy X Time. Note that 'reward' was not used as a demixing category (or task variable) in the algorithmic implementation since the influences of rewards could be detected from the firing patterns in demixed decision and strategies components itself. Additionally, we used a regularisation term in dPCA's cost function to prevent the model from overfitting.

## A.2 TASK SIMULATION AND TEMPORAL DIFFERENCE LEARNING

The task is simulated in 5 x 5 grid (Fig 8), with rewards being placed only at the terminal states. The simulations consider rule-switching and random start-location switching conditions as in the original experiment. However, the simulation design does not incorporate use of spatial cues for the allocentric task.

TD(0) and TD($\lambda$) algorithms are employed in the simulations to analyse the back propagation of estimated value and TD error from the rewarding states/time-points to the states/time-points coming before the reward states. We used two value functions at this stage for the two different strategies to see how each compares against the other. We also use eligibility traces to verify the notion of the hippocampus employing a decaying memory trace of any important spatiotemporal event. Eligibility traces enable the model to distribute credit efficiently to past states that can predict future rewards. These traces have been shown to augment learning in reinforcement learning models (Sutton & Barto (1998)).

Furthermore, an $\epsilon$ greedy online Q-learning algorithm to model policy learning (what actions to take when and where) of animals in the allocentric and egocentric tasks is implemented. We add to the $\epsilon$-greedy approach the following Q-value update function in Equation 1 to ensure rapid convergence of the model.

$$Q(S_t, A_t) = Q(S_t, A_t) + \alpha[R_{t+1} + \gamma * max_a Q(S_{t+1}, a) - Q(S_t, A_t)] \tag{1}$$

Here, $\alpha$ is the learning rate, $\gamma$ is discount factor and $Q(S_t, A_t)$ is the Q-value of a state-action pair at time t.

We think of this slight modification in the algorithm as combining best of the two worlds (SARSA and offline Q-learning) and this variant as twice-as-greedy online Q-learning. Nevertheless, we also verify our results from this modified model with a standard SARSA model to observe any discrepancies in learning policies, which we do not find. Additionally, this model does seem to converge slightly faster than a standard SARSA model(performance metrics not shown).

## A.3 SEPARATE DQN NETWORKS TO LEARN ALLOCENTRIC AND EGOCENTRIC STRATEGIES

We use deep Q neural networks to simulate the behavioural experiment. Figs 9A and 9B show the results from using two deep Q-networks to learn the allocentric and egocentric tasks separately. We include task-specific information into the input maze to assist the network in recognizing changing reward positions while following a single type of navigational rule. It seems that Deep Q neural networks can learn more efficiently than the Tabular Q-learning algorithms. Overall performance accuracies of both the networks are above chance levels, which shows that the network is able to cope with changing start and reward positions within a particular type of strategy (allocentric or egocentric). However, moving averages in the allocentric task show the occurrence of catastrophic forgetting as the network struggles to cope with changing reward locations while starting from the same start positions. On the other hand, the DQN that learns only the egocentric task performs perfectly after 130 trials and is even able to deal with a change in reward locations at trial 153 (dashed line). Changing reward locations within a task type are also denoted as strategy switches in the figure.

**A - Egocentric Spatial Rules**

| -1 | -1 | **-1** | -1 | -1 |
|----|----|----|----|----|
| -1 | -1 | **-1** | -1 | -1 |
| **0** | **0** | **0** | **0** | **1** |
| -1 | -1 | **0** | -1 | -1 |
| -1 | -1 | **0** | -1 | -1 |

(i) Start South

| -1 | -1 | **0** | -1 | -1 |
|----|----|----|----|----|
| -1 | -1 | **0** | -1 | -1 |
| **1** | **0** | **0** | **0** | **0** |
| -1 | -1 | **-1** | -1 | -1 |
| -1 | -1 | **-1** | -1 | -1 |

(ii) Start North

| -1 | -1 | **0** | -1 | -1 |
|----|----|----|----|----|
| -1 | -1 | **0** | -1 | -1 |
| **0** | **0** | **0** | **0** | **1** |
| -1 | -1 | **-1** | -1 | -1 |
| -1 | -1 | **-1** | -1 | -1 |

(iii) Start North

| -1 | -1 | **-1** | -1 | -1 |
|----|----|----|----|----|
| -1 | -1 | **-1** | -1 | -1 |
| **1** | **0** | **0** | **0** | **0** |
| -1 | -1 | **0** | -1 | -1 |
| -1 | -1 | **0** | -1 | -1 |

(iv) Start South

**B - Allocentric Spatial Rules**

| -1 | -1 | **-1** | -1 | -1 |
|----|----|----|----|----|
| -1 | -1 | **-1** | -1 | -1 |
| **1** | **0** | **0** | **0** | **0** |
| -1 | -1 | **0** | -1 | -1 |
| -1 | -1 | **0** | -1 | -1 |

(i) Start South

| -1 | -1 | **0** | -1 | -1 |
|----|----|----|----|----|
| -1 | -1 | **0** | -1 | -1 |
| **1** | **0** | **0** | **0** | **0** |
| -1 | -1 | **-1** | -1 | -1 |
| -1 | -1 | **-1** | -1 | -1 |

(ii) Start North

| -1 | -1 | **0** | -1 | -1 |
|----|----|----|----|----|
| -1 | -1 | **0** | -1 | -1 |
| **0** | **0** | **0** | **0** | **1** |
| -1 | -1 | **-1** | -1 | -1 |
| -1 | -1 | **-1** | -1 | -1 |

(iii) Start North

| -1 | -1 | **-1** | -1 | -1 |
|----|----|----|----|----|
| -1 | -1 | **-1** | -1 | -1 |
| **0** | **0** | **0** | **0** | **1** |
| -1 | -1 | **0** | -1 | -1 |
| -1 | -1 | **0** | -1 | -1 |

(iv) Start South

Figure 8: T-maze layout in a 5 x 5 matrix grid for the online Q-learning simulation. Grid layouts mimic the original allocentric and egocentric rules in the behavioural experiment. The value -1 denotes un-walkable cells because it is an elevated plus maze (convertible to a T-maze). The value 0 denotes walkable cells. The middle row has a 1 at either of its extremes to represent a reward. Ideal trajectories that should be chosen by the animal/agent are marked with color-coded arrows.

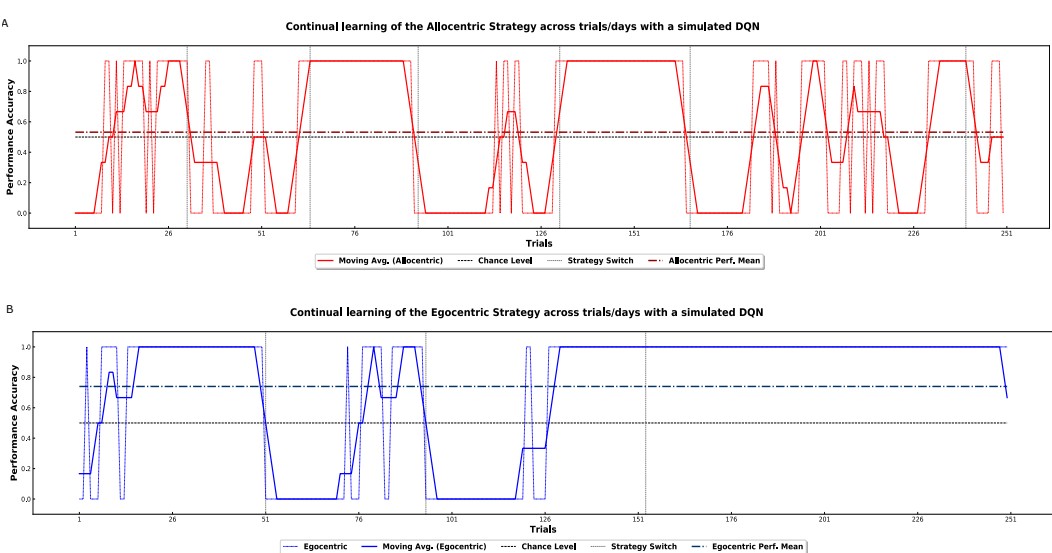

Figure 9: Task switching simulation using two separate deep Q networks. Total number of trials over which the networks learn the tasks independently is 250. Solid darker lines represent moving averages. Dashed lines denote change of reward locations during a type of task and can be called a strategy switch.(A) Performance accuracy of a DQN learning the allocentric task. (B) Performance accuracy of a DQN learning the egocentric task.

