# OpenReview forum: "HIPPOCAMPAL NEURONAL REPRESENTATIONS IN CONTINUAL LEARNING"
_ICLR.cc/2020/Conference — Reject_

### Official Review · AnonReviewer2 · 2019-10-21
**Official Blind Review #2**

**Rating:** 1

**Review:**

This paper analyses a dataset of representations in the CA1 region of the hippocampus of a rat conducting a spatial plus maze task that switches between allocentric and egocentric versions. In the allocentric version of the task, the rat must always go from north or south arms to the west arm to receive a reward. In the egocentric version the rat must always turn right, regardless of whether they are starting in the north or south arm. These two versions of the task are switched after some period of time.

The researchers conducted demixed principal components analysis (dPCA) on the data, then compared the activity of the components to aspects of a tabular Q-learner. They argue that some of the components match variables used by the Q-learner. Following this, they examined the ability of both the tabular Q-learner and a deep Q-network to perform this task, and compared their performance to the rat. They find that the rat is better than the Q-learners at continually updating from the egocentric to allocentric task, and vice-versa.

The goals of this paper are fantastic. I really like the attempt to link hippocampal activity with RL representations. But, this study is very muddled, and there are very serious problems with the paper that render it inappropriate for acceptance to ICLR. The five most major issues are:

(1) The choice of the demixing categories is missing a crucial category: space! Given the importance of the hippocampus for spatial representations, surely a big component of the variance can be explained by the animal's location in space. Why not include this? It might in fact be that some of the "time" components are just reflecting the usual spatial location of the animal during different components of the task.

(2) The identification of components from the neural data with things like reward prediction errors, eligibility traces, etc., is all done in a qualitative manner with no statistical controls. This lack of quantitative assessment for some of the key claims in the paper is just not okay for an academic paper. Moreover, even the qualitative claims are underwhelming. The authors' claim, for example, that time component #1 is an eligibility trace is a real stretch, in my opinion (see more below in point 3).

(3) The comparison between the animal data and the Q-learner is also done in a qualitative fashion that was extremely underwhelming. Fig 4 A & B were the most egregious. Are the authors really attempting to claim that the curves in 4A are clearly related to the curves in 4B? That's a real stretch, and to provide no quantitative assessment of this claim renders it completely unconvincing.

(4) The fact that model-free reinforcement learning algorithms cannot adapt in changing environments/tasks has been known for a long time. As such, the result showing that the Q-learners cannot switch easily between the tasks is not novel. See, for example, this paper: https://www.sciencedirect.com/science/article/pii/S0896627313008052

(5) Even if we accept the central claims from this paper, there is very little provided for machine learning researchers at ICLR to benefit from. What about the hippocampal representations makes the rats better at continual learning? What inductive biases or memory mechanisms might we glean from this work? Nothing like that is provided. As it stands, even being charitable, this paper really only speaks to a neuroscience audience, since even the Q-learning components are used only to understand the neuroscience data, not to think about how this could inform new ML systems or theories.

In addition to these problems, there are several small ones:

- Was only one animal included in this analysis? That's never stated, but Figs. 5 & 7 seem to suggest that. Not only does this 100% need to be stated, but it's very problematic from a generality standpoint.

- What type of recordings were these? How were individual cells identified (e.g. spike sorting)?

- How was the Deep Q-net trained exactly? The authors say it was trained on the first 250 trials from the animal, but that's confusing? Was it not trained to perform the task itself? Also, where is all the info on memory buffers, hyperparameters, etc. There is no way to reproduce these simulations given the lack of detail here.

- Some of the plots are confusing and hard to follow. For example, in figure 7 B and C, what determines the X-axis? What should I be looking for in the curves?

**Experience Assessment:**

I have published in this field for several years.

**Review Assessment: Checking Correctness Of Derivations And Theory:**

N/A

**Review Assessment: Checking Correctness Of Experiments:**

I assessed the sensibility of the experiments.

**Review Assessment: Thoroughness In Paper Reading:**

I read the paper at least twice and used my best judgement in assessing the paper.

---

> ### Author Response · Authors · 2019-11-15
> **Replies to reviewer comments**
>
>  We would like to thank the reviewer for the thoughtful comments. Our replies to the major issues pointed out are as below:
>
> (1) We agree that space is crucial when considering the hippocampus and this is very much an element that we are currently investigating. In our present analysis we only considered activity around the reward zone 2 seconds before and 4 seconds after entering the reward zone (reward sensor point). During this time we think that the importance of any spatial component is reduced as this is only around the reward zone.
>
> (2) We agree with the reviewer that our initial results need further analysis. However, we should note that the components found were consistently found across different animals, and changed with learning in a way consistent with theoretical ideas. We are in the process of further testing the statistical relevance of our results.
>
> (3) Similar to before, we agree that a closer comparison between RL methods and our observations is needed, and we are working on this. We thought that it would be of interest to the community to share our initial results.
>
> (4) The reviewer is right that model-free reinforcement learning models suffer from catastrophic forgetting and fail to adapt to changing environments. However, to our knowledge no previous attempts have been made to directly compare animal behaviour with RL algorithms in a continual learning setting. We are using this as a framework to test different algorithms. We are currently testing recently proposed continual learning algorithms, which are less prone to the catastrophic forgetting problem and are very much interested in comparing these methods with our observations both behaviourally and in terms of the representations learnt.
>
> (5) We agree that the current focus of the paper is towards understanding how biological systems solve these tasks. However, if one can provide insights into what elements seem to be relevant in neuroscience, these are also likely to be of relevance for machine learning neural networks. More generally, this can be seem as a framework to contrast biological and artificial algorithms, to identify similarities and differences, which may enable the field to identify new solutions. In a following version of this work, we will try to highlight insights that may be useful for machine learners.
>
> Our replies to the additional smaller problems reported by the reviewer:
>
> - Neural activity data from four Long Evans rats were used for the dPCA analysis. However, for the behavioural experiment simulation, we only use behavioural patterns in terms of which trials were allocentric and egocentric during the actual experiment of a single animal SC03. The results are consistent across animals, but we did not have the space to show all these results.
>
> - We should have clarified this. The neural recordings were obtained using multiple tetrodes in dorsal and ventral CA1 hippocampus. A standard spike sorting mechanism was used as described in Ciocchi et al. 2015, which for our analysis were converted into firing rates.
>
> - The first 250 trials from the animal contained the trial blocks for allocentric and egocentric tasks. Hence, the network was trained to perform the tasks using exactly the same experimental trials used in the animal. We also used a experience replay buffer. We will provide this details in a update version of the ms.
>
> - We will try to improve this in the next version. The x-axis shows the trial numbers from 4 trials before the allocentric task switches to an egocentric task at trial 111. In Fig 7B Moving averages of trial outcomes from the next 56 trials after task switch are plotted to show how fast the models or the animals learn to reach correct reward locations in the egocentric task. The colored dashed lines at trial number 106 to 110 show the average performance accuracy values from the end trials for the allocentric task. Similarly in Fig 7C, the relearning rates of the models and the animals after a task switch from an egocentric to an allocentric task is plotted against the trial number around which the task switch takes place. From these curves, we aim to show that an animal can almost immediately recognize that the task type has changed as indicated by performance higher than chance level just after a switch. RL models and neural networks, on the contrary, take significant number of trials to recognize the switch and relearn and act accordingly to the new task type. This points to the belief that the hippocampus CA1 perhaps stores information of allocentric and egocentric tasks separately and can fall back to either of those when required or there is a mechanism to consolidate information in a single neuronal cluster and leverage previously learned information to perform new tasks. Standard neural networks fail to maintain distinct values for the changing tasks even when trained over hundreds of trials.

---

### Official Review · AnonReviewer3 · 2019-10-23
**Official Blind Review #3**

**Rating:** 3

**Review:**

This work focuses on the problem of continual learning. It first proposes an analysis of neural recordings from rodents hippocampus that perform a task related to continual learning. The authors then claim to identify similarities in representations of behaviorally relevant variables between biological networks and standard artificial RL systems. Finally the authors propose a DQN implementation of the task reaching performance similar to rodents, although the implementation fail to perform continual learning.

I have found the results of the analysis of neural data interesting and well explained. However I haven’t been able to judge positively the main part of the manuscript pertaining to the comparison with artificial agents, maybe because of the lack of clarity in the exposition of the results (see detailed comments below). As such I think this manuscript is not ready yet for publication in ICLR.

Detailed comments:

Introduction:
The introduction reads well, it would be useful to have a short paragraph explaining the vast problem of continual learning, and situating the approach of the authors in this vast field. In particular it would be useful to explain in what sense the task considered is related to continual learning (switches between allocentric and egocentric tasks that are not informed by experimenter, but that the rodent has to figure out), because for now it is only briefly stated in the legend of figure 1.

Analysis of neural data:
Analysis of neural data using dPCA reveals interesting results regarding the representations of correct/incorrect trials, allocentric/egocentric tasks, temporal structure. Globally these results are well presented. Regarding figure 3, it would be nice to define what are early and late trials, is it a distinction inside a block where late trials are just before a switch. Or are late trials after weeks of training, in which case I do not really see the link with continual learning ?


Comparison between standard RL and hippocampal representations:

The idea to compare hippocampal representations and standard RL implementations is original and interesting. The exposition of the results, however, lacks important information for me to assess the relevance of these comparisons:

Fig4A(i): could the author explain what is on the x-axis ? What is the 1-D environment mentioned in the caption (in SM a 5*5 grid world is mentioned) ? Why not run TD on the same setting as Q-learning ? What is the black curve « value after learning » ? Why value depends on the x-axis ?
Fig4C: How are the two curves obtained ? As both tasks share the same Q function, it might be worth explaining how the two curves are obtained.

The authors mention that « policy information is override as starting location changes over » ? Could this fact be explained ? It would seem to me that starting points north s_north and south s_south would have distinct Q(s_north,.) and Q(s_south,.) avoiding overriding.

Also I am quite surprised that Q-learning fails on such a simple task (cf Fig5B), could the authors explain how many trials would be required to reach perfect performance by focusing on either of the two tasks ?

In this section a natural extension could be to use hierarchical RL, or options to model behavioral strategies such as allocentric and egocentric and compare with the hippocampal recordings.


Comparing DQN and animal performance:

A first crucial point to clarify is whether the DQN receives information about what task to perform (it is mentioned: « task-specific information provided to the neural network as input«, «this network with task-specific memory«, then a DQN « without task-specific information » is mentioned). This seems rather crucial, because if information about allocentric VS egocentric task is provided, contrary to the behavioral task for rats, then the network is asked to do strategy switches based on some cues, which is a different problem than continual learning.

If the DQN receives information about whether it is in a allocentric or egocentric trial, I am very surprised it is not able to perform this simple task.
Could the authors explain why the DQN has similar performance as the animal while it is shown to relearn slower than animals (is it because it learns better before task switch).
In figure 7, the curves for the model are very noisy and hence difficult to interpret, it would be nice to show averages over many same task switches, to get a clearer picture.

**Experience Assessment:**

I have read many papers in this area.

**Review Assessment: Checking Correctness Of Derivations And Theory:**

I assessed the sensibility of the derivations and theory.

**Review Assessment: Checking Correctness Of Experiments:**

I assessed the sensibility of the experiments.

**Review Assessment: Thoroughness In Paper Reading:**

I read the paper thoroughly.

---

> ### Author Response · Authors · 2019-11-14
> **Replies inline with the reviewer comments**
>
> We would like to thank the reviewer for the helpful comments.  Our comments are under the headings as in the detailed comments section.
>
> Introduction:
> Reply: We agree that the continual learning aspects of the task should be more clearly highlighted as this is central to our study. We will clarify this in the next version of the article.
>
> Analysis of neural data:
> Reply: We thank the reviewer for pointing out this lack of clarity, which is partially due to space constrains. Early trials are in the initial stages of learning, while the late trials correspond to after a few days of training.
>
> Comparison between standard RL and hippocampal representations:
> Reply to comments of Fig 4(A)i: Thank you for highlighting this, we agree that some of the elements need a better description. In Fig 4A(i), the x-axis corresponds to discretized timesteps. In this panel we are showing the elements of a simple TD algorithm in a 1D environment, so the x-axis represents timesteps in this 1d environment -- each timestep is also considered to be a new state. This simple setup helps us to demonstrate the key properties (eligibility trace and value function in this case) of a TD model in the presence of positive reinforcement, which could then be compared with the CA1 neural activity patterns. The black curve represents the value fuction over time (each timestep is a different state). We do agree that it would have been better to use the same algorithmic framework throughout (so similar to Q-learning as used in C) and we are currently working on this. The reason for having TD-learning first in 1D environment and then 2D, was to try to keep the model as simple as possible.
>
> Reply to comments on Fig 4C: We are going to clarify these points in a next version of the ms. In Fig 4C, the lines are obtained by taking the average of Q-values from allocentric correct/incorrect and egocentric correct/incorrect trials. These average Q-values are then plotted against the four timesteps required to reach the reward/no reward terminal states in the 5x5 grid. Calculated average values from allocentric and egocentric trials turn out to be distinct and hence the separation between the lines.
>
> Reply to comments on policy information: The starting points have distinct Q-values for the north and south points. However, since the experiment involves allocentric and egocentric tasks where both tasks can start from either north or south, a switch in the task overwrites the Q-values of the start locations pertaining to the previous task. Hence, the model forgets the Q-values of north and south start points learnt for an allocentric task when starts performing an egocentric task from the same start points. We will try to clarify this.
>
> Reply to comments of Q-learning: The Q-learning models (DQN and Tab-Q) take about 50 trials with changing start locations an allocentric task and around 65 for an egocentric task to converge to perfect performance. However, when we have these strategies interleaved between each other, the agent finds it difficult to learn both tasks one after another and we can see catastrophic interference.
>
> Reply to suggestions on using hierarchical RL: Recently, we have implemented a DQN with Elastic Weight Consolidation as in the paper by Kirkpatrick et al 2017. This model should be able to deal to perform continual learning and model the behavioural experiment with switching allocentric and egocentric tasks. We are currently analysing the results of this model.
>
> Comparing DQN and animal performance:
> Reply to comments on DQN: We first tested the network without task-specific information, but in this case the network performance is even worse. We added task-specific information to see whether the network could perform the task. We agree that this is a close representation of the experimental setup, but this further suggests that some other element is needed for these networks to be able to solve continual learning, as not even task-specific information appears to be  sufficient. In a future version we will include both models (with and without task-specific information).
>
> Reply to DQN animal comparison: This boils down to the catastrophic interference issue in systems such as neural networks. As the reviewer mentions the network can indeed learn quicker each task, but fails to retain any information about previous tasks.
>
> Reply to comments for Fig 7: Thanks for this suggestions. We are already working on this.

---

### Official Review · AnonReviewer1 · 2019-10-24
**Official Blind Review #1**

**Rating:** 6

**Review:**

This paper describes an analysis of hippocampal neuronal activity in rodents during spatial navigation tasks. Features were extracted from the data using a component analysis technique, and these features were then compared to quantities which arise during training of the DQN reinforcement learning algorithm.

Unfortunately I am not well versed enough in this literature to assess the merits of this submission. However, I could not find any glaring issues, unclear sentences or any other obvious sign of incompetence.

My apologies for the inadequacy or this review (and/or the paper assignment algorithm).

**Experience Assessment:**

I do not know much about this area.

**Review Assessment: Checking Correctness Of Derivations And Theory:**

I assessed the sensibility of the derivations and theory.

**Review Assessment: Checking Correctness Of Experiments:**

I assessed the sensibility of the experiments.

**Review Assessment: Thoroughness In Paper Reading:**

I made a quick assessment of this paper.

---

> ### Author Response · Authors · 2019-11-15
> **Thanking the reviewer**
>
> We thank you for your time and comments.

---

### Official Review · AnonReviewer4 · 2019-11-13
**Official Blind Review #4**

**Rating:** 1

**Review:**

This work sits at the intersection between neuroscience and machine learning. It proposes to use neural recordings from the rodent hippocampus to shed light into how biological agents achieve continual learning. The approach involves (i) analyzing neural data from the rodent hippocampus with the goal of identifying how neurons encode various behavioral variables; (ii) training different RL agents to solve a computer version of the same task, including tabular and DQN agents; (iii) contrasting the performance and representations of the artificial agents with those of animals.

The paper has an admirable goal: to find links between neuroscience and machine learning, using tools from one to promote advances in the other. When done correctly, this interaction can be fruitful and produce incredible insights for both fields. However, I believe that this paper does not deliver on either end due to major methodological and analytical flaws, rendering it innapropriate for ICLR. Below I list my major critiques:

1. The interpretations of the dPCA components seems very preliminary and lacks methological rigor. In particular, many alternative interpretations of the components are possible. For instance: Component #3 might represent outcome (rather than decision); Component #4 might represent greater engagement of the hippocampus in the egocentric task (e.g. Packard and McGaugh, 1996); etc. Moreover, the reported analyses do not include animal position, which will almost certainly be a major driver of population activity in the hippocampus (see work on place cells). Since the animal position is not reported, it is impossible to know if the dPCA components (e.g. Component #2) are, instead, representing alocentric position. Including animal position (perhaps also as an explanatory variable in the dPCA) might help, but the authors should also do a more thorough job testing their specific hypotheses beyond simply reporting them based on visual inspection.

2. The comparison between early and later training sessions is also rather crude and qualitative. The authors makes several claims about differences that are not tested directly with a proper hypothesis test.

3. The comparison between RL quantities and dPCA component is also very weak. For instance, the claim that a given dPCA component represents an eligibility trace needs much more evidence than simply showing that this component decays over space when the eligibility trace also decays. With respect to Component #4 (a critical component presumably related to the authors' hypothesis of multi-task representation), the authors report that, for an e-greedy agent, the average value is higher for alocentric vs. egocentric task. Yet, this difference is not investigated further and, again the authors claim victory based on a simple visual comparison between two plots (4C vs. 4D).

4. For most of the paper, the authors report the results from a single (typical?) animal. Ideally, the results for all animals would be reported (or some statistically sound aggregate of all animals).

5. Finally, the authors compare the performance of animal 3 with the performance of different RL agents. Again, this comparison is incredibly superficial and neglects many confound variables. The fact that the accuracy of both the animal and DQN is around 70% is not sufficient to claim that DQN is a good model for the animal's behavior, or that it is superior to other models that achieve slightly worse or better performance. Much more work needs to be done to properly compare RL and animals (e.g. comparing, in addition to performance, representations across various layers, prediction error signals, reward signals, etc).

Overall, while I commend the authors for an intriguing idea, the execution of the idea is so poor that I consider the paper to be of little interest to either the neuroscience or the machine learning communities. Therefore, I cannot recommend the paper for publication at ICLR.

**Experience Assessment:**

I have published one or two papers in this area.

**Review Assessment: Checking Correctness Of Derivations And Theory:**

N/A

**Review Assessment: Checking Correctness Of Experiments:**

I assessed the sensibility of the experiments.

**Review Assessment: Thoroughness In Paper Reading:**

I read the paper at least twice and used my best judgement in assessing the paper.

---

> ### Author Response · Authors · 2019-11-15
> **Replies to the reviewer comments**
>
> We thank the reviewer for the thoughtful comments. Our replies to the major critiques are as below:
>
> 1. We agree that space is likely to be crucial here, and are in the process of including this in the analysis and see how this changes our components. We have tried to reduce the importance of space by only considering activity around the reward zone.
>
> 2. We have decided to include our initial analysis of this results, but are now working on quantifying this more closely.
>
> 3. We thank the reviewer for this comment. But we should note that we only claim that our visual inspection is suggestive, far from conclusive. We are currently doing a quantitative comparison.
>
> 4. The dPCA results are based on recordings from hippocampus CA1 neurons of all four rodents available. For the continual learning simulation, we use the behaviour of one typical animal SC03 to train and test the Q-learning models (Tab-Q and DQN). We took this approach because of space constraints and also due to the stereotypical behaviour across the four different animals. However, we are currently working on including the continual learning results from all animals and agents.
>
> 5. Thank you for the comments. We should point out that we do not think that the DQN is a good model. The point we are making is that even though it achieves a relatively good average performance it completely fails in remembering the previous task (i.e. fails in continual learning). We are now implementing a DQN with elastic weight consolidation, which was designed to deal with continual learning problems. Next, we will be comparing the representations of the different methods in a more quantitative/precise way.

---

### Decision · Program_Chairs · 2019-12-19

**Decision:**

Reject

**Comment:**

This paper analyzes neural recording data taken from rodents performing a continual learning task using demixed principal component analysis, and aims to find representations for behaviorally relevant variables. They compare these features with those of a deep RL agent.

I am a big fan of papers like this that try to bridge between neuroscience and machine learning. It seems to have a great motivation and there are some interesting results presented. However the reviewers pointed out many issues that lead me to believe this work is not quite ready for publication. In particular, not considering space when analyzing hippocampal rodent data, as R2 points out, seems to be a major oversight. In addition, the sample size is incredibly small (5 rats, only 1 of which was used for the continual learning simulation). This seems to me like more of an exploratory, pilot study than a full experiment that is ready for publication, and therefore I am unfortunately recommending reject.

Reviewer comments were very thorough and on point. Sounds like the authors are already working on the next version of the paper with these points in mind, so I look forward to it.